# A new generator for proposing flexible lifetime distributions and its properties

**Muhammad Aslam** [1]*, **Christophe Ley**[2], **Zawar Hussain**[3], **Said Farooq Shah** [4], **Zahid Asghar**[1]

**1** Quaid-i-Azam University, Islamabad, Pakistan, **2** Department of Applied Mathematics, Computer Science and Statistics, Gent University, Ghent, Belgium, **3** Department of Social & Allied Sciences, Cholistan University of Veterinary & Animal University, Bahwalpur, Pakistan, **4** Department of Statistics, University of Peshawar, Peshawar, Pakistan

* maslam@stat.qau.edu.pk

**Data Availability Statement:** All relevant data are within the paper and its Supporting Information files.

**Funding:** The author(s) received no specific funding for this work.

## Abstract

In this paper, we develop a generator to propose new continuous lifetime distributions. Thanks to a simple transformation involving one additional parameter, every existing lifetime distribution can be rendered more flexible with our construction. We derive stochastic properties of our models, and explain how to estimate their parameters by means of maximum likelihood for complete and censored data, where we focus, in particular, on Type-II, Type-I and random censoring. A Monte Carlo simulation study reveals that the estimators are consistent. To emphasize the suitability of the proposed generator in practice, the two-parameter Fréchet distribution is taken as baseline distribution. Three real life applications are carried out to check the suitability of our new approach, and it is shown that our extension of the Fréchet distribution outperforms existing extensions available in the literature.

## Introduction

The modeling and analysis of lifetime phenomena is an important aspect of statistical work in a wide variety of scientific and technological fields. The field of lifetime data analysis has grown and expanded rapidly with respect to methodology, theory, and fields of application. In the context of modeling the real life phenomena, continuous probability distributions and many generalization or transformation methods have been proposed. These generalizations, obtained either by adding one or more shape parameters or by changing the functional form of the distribution, increase the flexibility of the distributions and model the phenomena more accurately. Extensive developments in software have made it possible to focus less on computational details and hence simplified the methods of estimation.

The following are prominent and highly cited generators or transformations proposed over the past years in the statistical literature for modeling lifetime distributions. [1] transform the survival function by adding an extra shape parameter. The exponentiated family of distributions, which adds a shape parameter as exponent to an existing cumulative distribution function (cdf), is presented by [2]. The beta-generated family by [3] is based on both Beta type-I and Beta type-II distributions, while the Kumaraswamy-generated family by [4] uses the

**Competing interests:** The authors have declared that no competing interests exist.

Kumaraswamy distribution instead of the Beta distribution. [5] pioneered a versatile and flexible gamma-G class of distributions based on the Generalized Gamma distribution.

Let $F(x; \zeta)$ be the cdf of a given random variable depending on some real-valued parameter (s) $\zeta$. Our approach in this paper consists in enriching this cdf by transforming it into

$$G(x; \xi) = \frac{\log \left\{ 2 - e^{-\lambda F(x;\zeta)} \right\}}{\log \left\{ 2 - e^{-\lambda} \right\}}, \tag{1}$$

where $\xi = (\lambda, \zeta)$ for some positive real-valued shape parameter $\lambda$ and the parameter $\zeta$ from the baseline distribution. We call this transformation the log-expo transformation (*LET*). It is aspired from [6] who considered the less versatile transformation

$$G(x; \xi) = 1 - \frac{\log \left\{ 2 - F(x; \zeta) \right\}}{\log 2}. \tag{2}$$

While their approach only allows modulating the shape of distributions in a fixed way, ours is more flexible since it contains the extra shape parameter $\lambda$ to regulate the transformation. To evaluate the suitability of the new proposed transformation, we will take the Fréchet distribution by [7] as example of baseline distribution throughout the rest of this paper.

The remaining paper is organized in the following order. The density function of the proposed method is defined and its basic statistical properties are derived. Next, we discuss, parameter estimation via maximum likelihood for complete and censored data, together with submodel likelihood ratio test. Monte Carlo simulation study to show the consistency of our estimation procedures. The fitting abilities of our new approach is illustrated by means of three real data sets. Finally, we give concluding remarks, and the Appendix collects densities of distributions used in the real data analysis.

## The proposed density and its properties

The probability density function (pdf) corresponding to Eq (1) is given by

$$g(x; \xi) = \frac{\lambda f(x; \zeta) e^{-\lambda F(x;\zeta)}}{\log \left\{ 2 - e^{-\lambda} \right\} \left\{ 2 - e^{-\lambda F(x;\zeta)} \right\}}, \; x \geq 0, \tag{3}$$

where $F(x; \zeta)$ and $f(x; \zeta)$ are the arbitrary cdf and pdf of the baseline distribution. The cdf and pdf given in Eqs (1) and (3), respectively, will be more readable for a given expression of $F(x; \zeta)$ and $f(x; \zeta)$ of any baseline distribution. The flexibility of the proposed family of distributions is increased by adding shape parameter $\lambda$. Hereafter, we say that the random variable $X$ having density Eq (3) is a log-expo transformed random variable.

The survival function $S(x; \xi) = 1 - G(x; \xi)$ is of the simple form $S(x; \xi) = 1 - \frac{\log \left\{ 2 - e^{-\lambda F(x;\zeta)} \right\}}{\log \left\{ 2 - e^{-\lambda} \right\}}$ and the hazard function $h(x; \xi) = \frac{g(x;\xi)}{S(x;\xi)}$ reads $h(x; \xi) = \frac{\lambda f(x) e^{-\lambda F(x;\zeta)}}{\left\{ 2 - e^{-\lambda F(x;\zeta)} \right\} \left[ \log \left\{ 2 - e^{-\lambda} \right\} - \log \left\{ 2 - e^{-\lambda F(x;\zeta)} \right\} \right]}$, and the reverse hazard function $h'(x; \xi) = \frac{g(x;\xi)}{G(x;\xi)}$ becomes $h'(x; \xi) = \frac{\lambda f(x;\zeta) e^{-\lambda F(x;\zeta)}}{\left\{ 2 - e^{-\lambda F(x;\zeta)} \right\} \log \left\{ 2 - e^{-\lambda F(x;\zeta)} \right\}}$. The explicit form of the $v^{th}$ quantile of the *LET* family of distributions is given by the simple expression $x = F^{-1} \left[ -\frac{1}{\lambda} \log \left\{ 2 - e^{u \log \left\{ 2 - e^{-\lambda} \right\}} \right\}; \zeta \right]$. Consequently, random number generation from the *LET* family of distributions turns out to be a straightforward task.

Now, for the sake of illustration, we will briefly present three submodels of the proposed family of distributions, based on the baseline Fréchet, Exponential and Lomax distributions.

**LET-Fréchet (*LET-F*) distribution.** Consider the Fréchet distribution with respective cdf and pdf $F(x; \alpha, \beta) = e^{-\left( \frac{\beta}{x} \right)^{\alpha}}$ and $f(x; \alpha, \beta) = \alpha \beta^{\alpha} x^{-\alpha-1} e^{-\left( \frac{\beta}{x} \right)^{\alpha}}$. The cdf and pdf of the

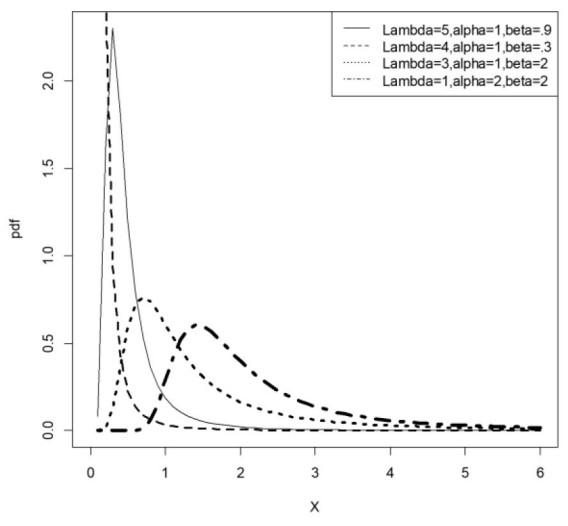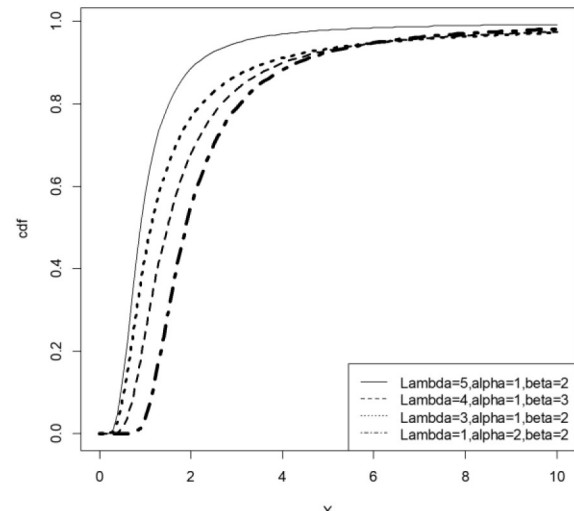

**Fig 1. Pdf and cdf plots of the *LET-F* distribution.**

LET-Fréchet distribution then correspond to

$$G(x; \lambda, \alpha, \beta) = [\log\{2 - e^{-\lambda}\}]^{-1}\log\left\{2 - e^{-\lambda e^{-\left(\frac{\beta}{x}\right)^{\alpha}}}\right\}, \lambda > 0, \alpha > 0, \beta > 0, \text{ and}$$

$g(x; \lambda, \alpha, \beta) = \dfrac{\lambda\alpha\beta^{\alpha}x^{-\alpha-1}e^{-\left(\frac{\beta}{x}\right)^{\alpha}}e^{-\lambda e^{-\left(\frac{\beta}{x}\right)^{\alpha}}}}{\log\{2-e^{-\lambda}\}\left\{2-e^{-\lambda e^{-\left(\frac{\beta}{x}\right)^{\alpha}}}\right\}}$. Fig 1 illustrates the possible shapes of the pdf and cdf of

the *LET-F* distribution.

Since the *LET-F* distribution is our red thread example, we also provide some moment expressions. In Table 1, we give the first four moments $u'_n$, $n = 1,\ldots,4$, the standard deviation (*SD*), coefficient of skewness (*CS*) and coefficient of kurtosis (*CK*) for different combinations of parameters. These values are calculated via Mathematica.

**LET-Exponential (*LET-E*) distribution.** Consider the Exponential distribution with respective cdf and pdf $F(x; \alpha) = 1 - e^{-\alpha x}$ and $F(x; \alpha) = \alpha e^{-\alpha x}$. The cdf and pdf of the LET-Exponential distribution then correspond to $G(x; \lambda, \alpha) = [\log\{2 - e^{-\lambda}\}]^{-1}\log\{2 - e^{-\lambda(1-e^{-\alpha x})}\}, \lambda > 0, \alpha > 0$, and $g(x; \lambda, \alpha) = \frac{\lambda\alpha e^{-\alpha x}e^{-\lambda(1-e^{-\alpha x})}}{\log\{2-e^{-\lambda}\}\{2-e^{-\lambda(1-e^{-\alpha x})}\}}$. Fig 2 illustrates the possible shapes of the pdf and cdf of the *LET-E* distribution.

**LET-Lomax (*LET-L*) distribution.** Consider the Lomax distribution with respective cdf and pdf $F(x; \alpha, \beta) = 1 - (1 + \alpha x)^{-\beta}$ and $f(x; \alpha, \beta) = \alpha\beta(1 + \alpha x)^{-\beta-1}$. The cdf and pdf of the LET-Lomax distribution then correspond to $G(x; \lambda, \alpha, \beta) = [\log\{2 - e^{-\lambda}\}]^{-1}\log$

$\{2 - e^{-\lambda(1-(1+\alpha x)^{-\beta})}\}, \lambda > 0, \alpha > 0, \beta > 0$ and $g(x; \lambda, \alpha, \beta) = \frac{\lambda\alpha\beta(1+\alpha x)^{-\beta-1}e^{-\lambda(1-(1+\alpha x)^{-\beta})}}{\log\{2-e^{-\lambda}\}\{2-e^{-\lambda(1-(1+\alpha x)^{-\beta})}\}}$. Fig 3 illustrates the possible shapes of the pdf and cdf of the *LET-L* distribution.

## Lifetime data analysis and parameter estimation

The data encountered in survival analysis and reliability studies are often censored. This is why, besides classical maximum likelihood estimation, we also show how to estimate the parameters of our new family of distributions when the data are censored. More precisely, we consider Type-II, Type-I and random (right) censoring. These censoring schemes have been

**Table 1. Moments of the *LET-F* model for combinations of parameters.**

|  | $\lambda = 3, \alpha = 6, \beta = 2$ | $\lambda = 3, \alpha = 6, \beta = 1$ | $\lambda = 3, \alpha = 5, \beta = 2$ | $\lambda = 5, \alpha = 6, \beta = 2$ | $\lambda = 5, \alpha = 6, \beta = 3$ |
|---|---|---|---|---|---|
| $u'_1$ | 1.8825 | 0.9413 | 1.8655 | 1.7847 | 2.6770 |
| $u'_2$ | 3.6429 | 0.9107 | 3.6324 | 3.2324 | 7.2729 |
| $u'_3$ | 7.3439 | 0.9180 | 7.6159 | 5.9719 | 20.1550 |
| $u'_4$ | 15.9440 | 0.9965 | 19.3117 | 11.3993 | 57.7088 |
| SD | 0.0988 | 0.0247 | 0.1522 | 0.0473 | 0.1065 |
| CS | 3.6626 | 3.6627 | 4.5784 | 3.3122 | 3.3122 |
| CK | 43.6080 | 43.6086 | 86.1270 | 47.6170 | 47.6170 |

employed in numerous fields, especially for crash rates on roads which are based on censored data. Such data can be handled by using tobit, multinomial logit, mixed logit, ordered logit probit/logit models. for example, see the articles [8–15]. Finally, we develop likelihood ratio tests for testing the suitability of the baseline distributions against our *LET* extension.

## Maximum likelihood estimation

We derive sample estimates of the unknown parameters of the *LET* model by using the maximum likelihood estimation technique. Let $x_1, x_2, \ldots, x_n$ be the observations of a random sample of size $n$ from the *LET* model. The likelihood function is given by

$$L(\xi) = \prod_{i=1}^{n}\left[\frac{\lambda f(x_i;\zeta)e^{-\lambda F(x_i;\zeta)}}{\log\{2 - e^{-\lambda}\}\{2 - e^{-\lambda F(x_i;\zeta)}\}}\right],$$

and the log-likelihood function by

$$ll(\xi) = n\log\lambda + \sum_{i=1}^{n}\log\{f(x_i;\zeta)\} - \lambda\sum_{i=1}^{n}F(x_i;\zeta) - n\log[\log\{2 - e^{-\lambda}\}]$$

$$- \sum_{i=1}^{n}\log\{2 - e^{-\lambda F(x_i;\zeta)}\}.$$

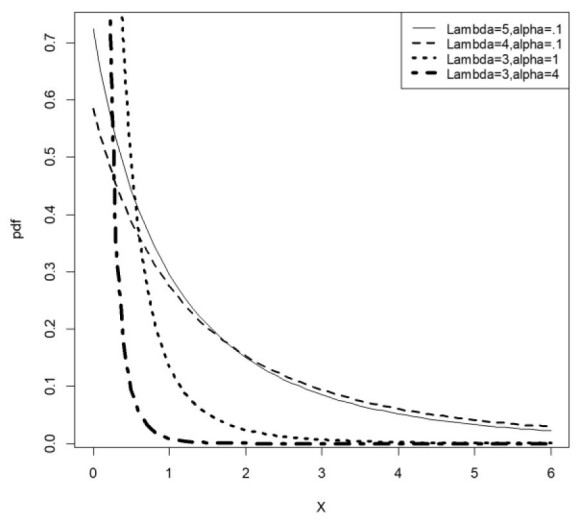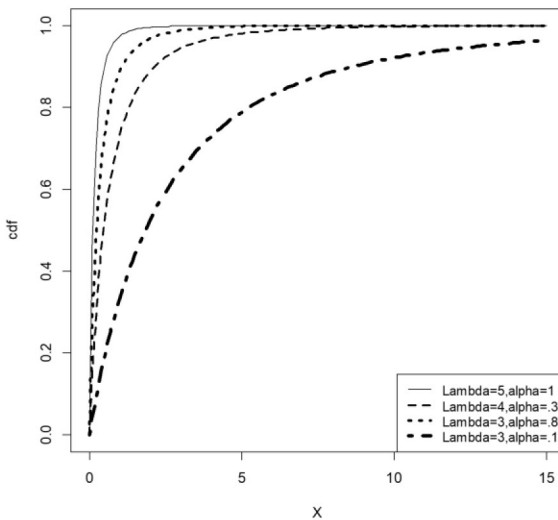

**Fig 2. Pdf and cdf plots of the *LET-E* distribution.**

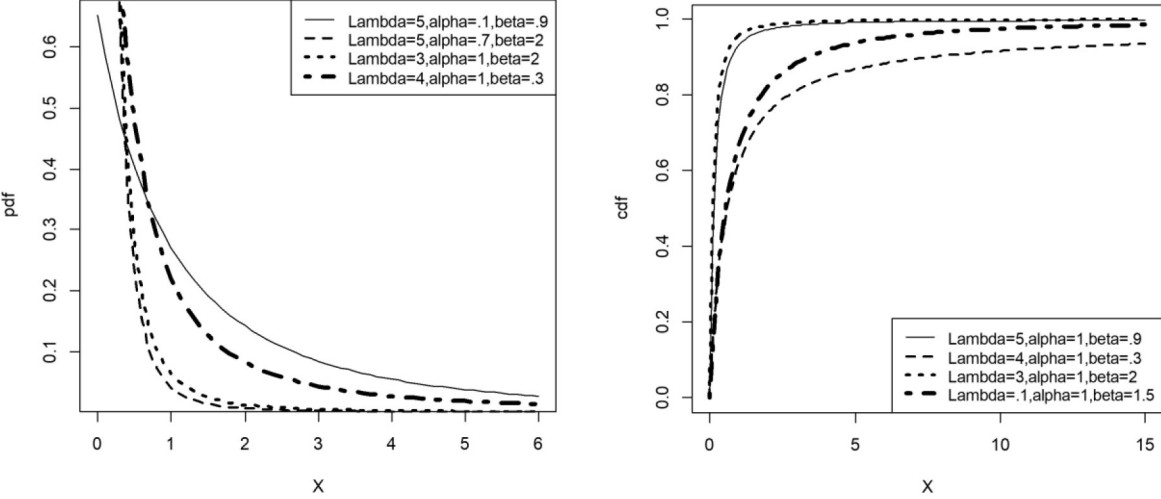

**Fig 3. Pdf and cdf plots of the *LET-L* distribution.**

Differentiating the log-likelihood with respect to $\lambda$ and $\zeta$ and equating to zero, we get the score equations

$$\frac{\partial ll(\xi)}{\partial \lambda} = \frac{n}{\lambda} - \sum_{i=1}^{n} F(x_i; \zeta) - \frac{ne^{-\lambda}}{\{2 - e^{-\lambda}\} \log \{2 - e^{-\lambda}\}} - \sum_{i=1}^{n} \left( \frac{F(x_i; \zeta) e^{-\lambda F(x_i; \zeta)}}{2 - e^{-\lambda F(x_i; \zeta)}} \right) = 0, \qquad (4)$$

and

$$\frac{\partial ll(\xi)}{\partial \zeta} = \sum_{i=1}^{n} \left( \frac{f^{\zeta}(x_i; \zeta)}{f(x_i; \zeta)} \right) - \lambda \sum_{i=1}^{n} F^{\zeta}(x_i; \zeta) - \lambda \sum_{i=1}^{n} \left( \frac{F^{\zeta}(x_i; \zeta) e^{-\lambda F(x_i; \zeta)}}{2 - e^{-\lambda F(x_i; \zeta)}} \right) = 0, \qquad (5)$$

where $f^{\zeta}(x_i; \zeta) = \frac{df(x_i; \zeta)}{d\zeta}$ and $F^{\zeta}(x_i; \zeta) = \frac{dF(x_i; \zeta)}{d\zeta}$. Solving Eqs (4) and (5) gives the maximum likelihood estimates of the unknown parameters $\lambda$ and $\zeta$. Typically, this requires numerical optimization techniques such as Newton-Raphson methods as given in [16 and 17].

## Parameter estimation under various types of right censoring

Let $x_1$, $x_2$, . . ., $x_n$ be the observations of a random sample of size $n$ from the *LET* model. In what follows, we explain how to perform maximum likelihood estimation in our *LET* model for three types of right censoring.

## Type-II censoring

In case of Type-II right censoring, $t$ observations out of the $n$ are censored from the right side. The likelihood function then becomes

$$L(\xi) = \frac{n!}{t!} [\prod_{i=1}^{n-t} g(x_{(i)}; \xi)][S(x_{(n-t)}; \xi)]^t,$$

where $x_{(i)}$ is the order statistic of order $i$, and the log-likelihood function, expressed in terms of

the original baseline distribution, reads

$$ll(\xi) = \log\left(\frac{n!}{t!}\right) + t\log\left[1 - \frac{\log\{2 - e^{-\lambda F(x_{(n-t)};\zeta)}\}}{\log\{2 - e^{-\lambda}\}}\right] + \sum_{i=1}^{n-t}\log\left[\frac{\lambda f(x_{(i)};\zeta)e^{-\lambda F(x_{(i)};\zeta)}}{\log\{2 - e^{-\lambda}\}\{2 - e^{-\lambda F(x_{(i)};\zeta)}\}}\right].$$

Differentiating this log-likelihood with respect to $\lambda$ and $\zeta$ yields the score equations

$$\frac{\partial ll(\xi)}{\partial \lambda} = \frac{n-t}{\lambda} - k_1(k_2 - k_3) - \sum_{i=1}^{n-t}\left[F(x_{(i)};\zeta) + \frac{e^{-\lambda}}{\{2-e^{-\lambda}\}\log\{2-e^{-\lambda}\}} + \frac{F(x_{(i)};\zeta)e^{-\lambda F(x_{(i)};\zeta)}}{2 - e^{-\lambda F(x_{(i)};\zeta)}}\right] = 0, \quad (6)$$

where $k_1 = \frac{t}{\log\{2-e^{-\lambda}\}[\log\{2-e^{-\lambda}\} - \log\{2-e^{-\lambda F(x_{(n-t)};\zeta)}\}]}$, $k_2 = \frac{\log\{2-e^{-\lambda}\}F(x_{(n-t)};\zeta)e^{-\lambda F(x_{(n-t)};\zeta)}}{2-e^{-\lambda F(x_{(n-t)};\zeta)}}$ and

$k_3 = \frac{\log\{2-e^{-\lambda F(x_{(n-t)};\zeta)}\}e^{-\lambda}}{2-e^{-\lambda}}$, and $\frac{\partial ll(\xi)}{\partial \zeta} = \frac{-t\lambda F^{\zeta}(x_{(n-t)};\zeta)e^{-\lambda F(x_{(n-t)};\zeta)}}{\{2-e^{-\lambda F(x_{(n-t)};\zeta)}\}[\log\{2-e^{-\lambda}\} - \log\{2-e^{-\lambda F(x_{(n-t)};\zeta)}\}]} + \sum_{i=1}^{n-t}\left[\frac{f^{\zeta}(x_{(i)};\zeta)}{f(x_{(i)};\zeta)}\right] -$

$$\lambda\sum_{i=1}^{n-t}\left[\frac{F^{\zeta}(x_{(i)};\zeta)}{F(x_{(i)};\zeta)}\right] - \lambda\sum_{i=1}^{n-t}\left[\frac{F^{\zeta}(x_{(i)};\zeta)e^{-\lambda F(x_{(i)};\zeta)}}{2 - e^{-\lambda F(x_{(i)};\zeta)}}\right] = 0. \quad (7)$$

Expressions (6) and (7) give the maximum likelihood estimates of the unknown parameters $\lambda$ and $\zeta$ for type-II right censored data. It is clear that their solution cannot be obtained analytically, and numerical techniques used in [16 & 17] are required.

## Type-I censoring

Suppose that a random sample of $n$ units from $G(x;\xi)$ is processed for a predefined time $x_c$ and then the process terminate. We observed the lifetime of $\delta$ observations before terminating the process and the remaining $n - \delta$ observations will be censored. Thus, the lifetimes are observed only if $x_i \leq x_c$ for $i = 1, 2, \ldots, n$.

Defining $I_i = \begin{cases} 1, & \text{if } X_i \leq x_c \\ 0, & \text{if } X_i > x_c \end{cases}$ and $\delta = \sum_{i=1}^{n} I_i$, the likelihood function can be written as

$L(\xi) = \left[\prod_{i=1}^{n} g(x_i;\xi)^{I_i}\right]S(x_c;\xi)^{n-\delta}$ and the log-likelihood function is given by

$$ll(\xi) = (n-\delta)\log\left[1 - \frac{\log\{2 - e^{-\lambda F(x_c;\zeta)}\}}{\log\{2 - e^{-\lambda}\}}\right] + \sum_{i=1}^{n} I_i \log\left[\frac{\lambda f(x_i;\zeta)e^{-\lambda F(x_i;\zeta)}}{\log\{2 - e^{-\lambda}\}\{2 - e^{-\lambda F(x_i;\zeta)}\}}\right].$$

The score equations, and associated maximum likelihood estimates, are obtained along the same lines as in the previous sections. Their solution cannot be obtained analytically, and numerical techniques given in [16 & 17] are required.

## Random censoring

Suppose a random sample consists of $n$ observations $T_1, T_2, \ldots, T_n$ from a continuous failure distribution $G(t;\xi)$ and consider other random censoring variables $C_1, C_2, \ldots, C_n$ drawn independently from a censoring distribution $H(c;\xi)$. The observations for right censored data are presented as $(X_i, I_i)$, $i = 1, 2, \ldots, n$, where $X_i = Min(T_i, C_i)$, and

$$I_i = \begin{cases} 1, & \text{if } T_i \leq C_i \\ 0, & \text{if } T_i > C_i. \end{cases}$$

The likelihood function for random censored data $x_1, x_2, \ldots, x_n$ can be written as

$L(\xi) = \prod_{i=1}^{n} g(x_i; \xi)^{I_i} S(x_i; \xi)^{1-I_i}$ which yields the log-likelihood function

$$ll(\xi) = \sum_{i=1}^{n} I_i \log \left[ \frac{\lambda f(x_i; \zeta) e^{-\lambda F(x_i; \zeta)}}{\log\{2 - e^{-\lambda}\}\{2 - e^{-\lambda F(x_i; \zeta)}\}} \right] + \sum_{i=1}^{n} (1 - I_i) \log \left\{ 1 - \frac{\log\{2 - e^{-\lambda F(x_i; \zeta)}\}}{\log\{2 - e^{-\lambda}\}} \right\}.$$

The score equations, and associated maximum likelihood estimates, are obtained along the same lines as in the previous sections. Their solution cannot be obtained analytically, and numerical techniques such as used in [16 & 17] are required.

## Submodel testing

Our *LET* extension paves the way for submodel testing of the baseline distribution by means of likelihood ratio tests. For each parameter $\xi$, we denote by $\hat{\xi}$ the unconstrained maximum likelihood estimate and by $\hat{\xi}_r$ the maximum likelihood estimate under the restricted submodel. For example, testing for the Fréchet distribution against the $LET - F$ model can be achieved by the test statistic $T_{Fréchet} = -2(ll(\hat{\alpha}_r, \hat{\beta}_r) - ll(\hat{\lambda}, \hat{\alpha}, \hat{\beta}))$, rejecting $H_0$: $\lambda = 0$ at asymptotic level $\alpha$ against $H_1$: $\lambda \neq 0$ whenever $T_{Fréchet}$ exceeds $\chi^2_{1;1-\alpha}$, the $\alpha$-upper quantile of the chi-squared distribution with one degree of freedom.

## Monte Carlo simulation results of the *LET-F* model

We perform a Monte Carlo simulation study in order to evaluate the behavior of maximum likelihood estimates of the proposed *LET-F* distribution for complete and censored data. The data were censored 10% from the right by using the Type-II and Type-I schemes. We calculate means, biases and mean-squared errors (*MSEs*) of each parameter of the *LET-F* model for different sample sizes *n*. To obtain the results, the process is replicated $N = 10{,}000$ times for $n = 20, 30, 50$ and $100$ for censored data, and we added the sample sizes 200 and 300 for the complete data. The simulated means, biases and *MSEs* for complete and censored data are provided in Tables 2 and 3, respectively. We observe that, overall, the estimation procedure works well and that the estimates become better with increasing sample size, as should be the case. It is noteworthy to remark that close-to-zero values of $\lambda$ are more difficult to estimate, which is probably due to the fact that such small values only slightly trigger our transformation as compared to the baseline model.

## Real data analysis

In this section, the fitting potential of our new procedure is evaluated by means of three real data sets, of which the last one is censored. In each case, we compare our *LET-F* model with competitors from the literature.

## Non-censored data

The first data set shows the failure stresses (in GPa) of 64 bundles of carbon fibres and is also used by [18]. The second data set is presented by [19] and concerns the survival time counted in days of guinea pigs with infected virulent tubercle bacilli.

The proposed *LET-F* model is compared with the basic Fréchet (*F*) distribution as well as other extensions of it, such as the logarithmic transformed Fréchet (*LTF*) of [6], the Exponentiated Fréchet (*EF*) as initiated by [2], the Marshall-Olkin Fréchet (*MOF*) of [1], and the Kumaraswamy Fréchet (*KF*) according to the construction of [4]. We use the Kolmogorov–Smirnov (*KS*), Cramer–von Mises (*W\**), Anderson-Darling (*A*) and Deviance Information

**Table 2. The simulated means, biases and *MSEs* of the *LET-F* model for complete data.**

| n | | $\alpha = 5$ | $\beta = 3$ | $\lambda = 0.2$ | $\alpha = 7$ | $\beta = 2$ | $\lambda = 1$ |
|---|---|---|---|---|---|---|---|
| 20 | Mean | 7.4202 | 7.4202 | 7.4202 | 6.1623 | 2.5125 | 44.1802 |
| | Bias | 2.4202 | 2.4202 | 2.4202 | -0.8377 | 0.5125 | 43.1802 |
| | MSE | 5.8575 | 5.8575 | 5.8575 | 0.7018 | 0.2626 | 1864.528 |
| 30 | Mean | 6.7879 | 3.0651 | 2.9379 | 6.3019 | 2.2937 | 13.1648 |
| | Bias | 1.7879 | 0.0651 | 2.7379 | -0.6981 | 0.2937 | 12.1648 |
| | MSE | 3.1967 | 0.0042 | 7.4960 | 0.4874 | 0.0863 | 147.981 |
| 50 | Mean | 5.9997 | 3.0253 | 0.9445 | 6.2938 | 2.1755 | 3.7590 |
| | Bias | 0.9997 | 0.0253 | 0.7445 | -0.7062 | 0.1755 | 2.7590 |
| | MSE | 0.9994 | 0.0006 | 0.5543 | 0.4987 | 0.0308 | 7.6118 |
| 100 | Mean | 5.3205 | 3.0377 | 0.5552 | 6.5032 | 2.1016 | 2.1361 |
| | Bias | 0.3205 | 0.0377 | 0.3552 | -0.4968 | 0.1016 | 1.1361 |
| | MSE | 0.1027 | 0.0014 | 0.1261 | 0.2468 | 0.0103 | 1.2907 |
| 200 | Mean | 5.0338 | 3.0558 | 0.5266 | 6.6136 | 2.0674 | 1.7273 |
| | Bias | 0.0338 | 0.0558 | 0.3266 | -0.3864 | 0.0674 | 0.7273 |
| | MSE | 0.0011 | 0.0031 | 0.1067 | 0.1493 | 0.0045 | 0.5290 |
| 300 | Mean | 4.9616 | 3.0579 | 0.4994 | 6.7025 | 2.0501 | 1.5386 |
| | Bias | -0.0384 | 0.0579 | 0.2994 | -0.2975 | 0.0501 | 0.5386 |
| | MSE | 0.0015 | 0.0033 | 0.0896 | 0.0885 | 0.0025 | 0.2901 |

Criterion (*DIC*), goodness-of-fit tests for the comparison. The *DIC* is a generalized form of *AIC* and is widely used for model adequacy (see, [20 and 21]). The best model exhibits the smallest value of these statistics. The results are obtained by using *R*. In the Appendix, we give the respective pdfs of the above mentioned distributions.

In Table 4, we provide the value of the *KS* test together with the related p-value. As we can see, our *LET-F* model exhibits twice the lowest *KS* value and, consequently, largest p-value. For the second data set, the *LTF* model (2), which we try to improve on in particular, is clearly rejected by the *KS* test statistic. To further corroborate the strength of our *LET-F* model, we provide in Table 5 the corresponding values of the $W^*$, the *DIC* and the *A* statistics. They also reveal that the *LET-F* model is very appropriate for these data sets as it outperforms its competitors. For the sake of illustration, the histogram of both data sets and fitted pdfs of all

**Table 3. The simulated means, biases and *MSEs* of the *LET-F* model under Type-II and Type-I censoring schemes.**

| | | Type-II (10%) | | | Type-I (10%) | | |
|---|---|---|---|---|---|---|---|
| n | | $\alpha = 4$ | $\beta = 1.5$ | $\lambda = 0.5$ | $\alpha = 5$ | $\beta = 3$ | $\lambda = 0.2$ |
| 20 | Mean | 7.1319 | 1.4222 | 1.3084 | 5.1853 | 3.2814 | 1.5743 |
| | Bias | 3.1319 | -0.0778 | 0.8084 | 0.1853 | 0.2814 | 1.3743 |
| | MSE | 9.8089 | 0.0060 | 0.6535 | 0.0344 | 0.0792 | 1.8886 |
| 30 | Mean | 6.0671 | 1.4452 | 0.4424 | 5.0811 | 3.2572 | 1.4502 |
| | Bias | 2.0671 | -0.0548 | -0.0576 | 0.0811 | 0.2572 | 1.2502 |
| | MSE | 4.2729 | 0.0030 | 0.0033 | 0.0066 | 0.0662 | 1.5631 |
| 50 | Mean | 5.0600 | 1.4898 | 0.4147 | 4.9503 | 3.2324 | 1.2883 |
| | Bias | 1.0600 | -0.0102 | -0.0853 | -0.0497 | 0.2324 | 1.0883 |
| | MSE | 1.1235 | 0.0001 | 0.0073 | 0.0025 | 0.0540 | 1.1843 |
| 100 | Mean | 3.9939 | 1.5588 | 0.5707 | 4.8597 | 3.1896 | 1.0618 |
| | Bias | -0.0061 | 0.0588 | 0.0707 | -0.1403 | 0.1896 | 0.8618 |
| | MSE | 0.0000 | 0.0035 | 0.0050 | 0.0197 | 0.0359 | 0.7428 |

**Table 4. KS and P-values of the considered models.**

| Data | Statistic | LET-F | LTF | EF | MOF | KF | F |
|---|---|---|---|---|---|---|---|
| 1 | KS | 0.0788 | 0.0972 | 0.0816 | 0.0827 | 0.0813 | 0.1006 |
| | P-Value | 0.8293 | 0.5908 | 0.7953 | 0.7823 | 0.7987 | 0.5471 |
| 2 | KS | 0.1007 | 0.2101 | 0.1225 | 0.1207 | 0.1031 | 0.1964 |
| | P-Value | 0.4582 | 0.0035 | 0.2297 | 0.2448 | 0.4283 | 0.0077 |

**Table 5. Cramer–von Mises ($W^*$), Anderson-Darling ($A$) and Deviance Information Criterion ($DIC$) values of the considered models.**

| Model | Statistics (Data set 1) | | | Statistics (Data set 2) | | |
|---|---|---|---|---|---|---|
| | $W^*$ | $A$ | $DIC$ | $W^*$ | $A$ | $DIC$ |
| LET-F | 0.0449 | 0.2697 | 121.9680 | 0.0936 | 0.6635 | 213.4680 |
| LTF | 0.1019 | 0.5681 | 130.9540 | 0.5152 | 3.2925 | 240.4150 |
| EF | 0.0619 | 0.3310 | 124.5210 | 0.1178 | 0.8483 | 213.5480 |
| MOF | 0.0736 | 0.3932 | 129.6140 | 0.0766 | 0.5873 | 223.6090 |
| KF | 0.0615 | 0.3299 | 122.3000 | 0.1156 | 0.8314 | 214.4140 |
| F | 0.1150 | 0.6420 | 134.1110 | 0.5261 | 3.3486 | 240.2510 |

considered models are provided in Fig 4, while Fig 5 exhibits the corresponding PP-Plots. The better fit of the *LET-F* model for both the data sets included in this study can thus also be recognized visually.

Finally, our likelihood ratio test yields a p-value of 0.027 for the first data set and 0.000 for the second data set. Thus, the Fréchet distribution is rejected in favour of the *LET-F* model for data set 2 at any level, while it is rejected at the classical 5% level for the first data set but no longer at, the 2% level. The maximum likelihood estimates (*MLEs*), Bayes estimates (*BEs*), and their corresponding standard errors (*SEs*) and posterior standard deviations (*SDs*), respectively, for the parameters of the *LET-F* and the competitor models are given in Table 6.

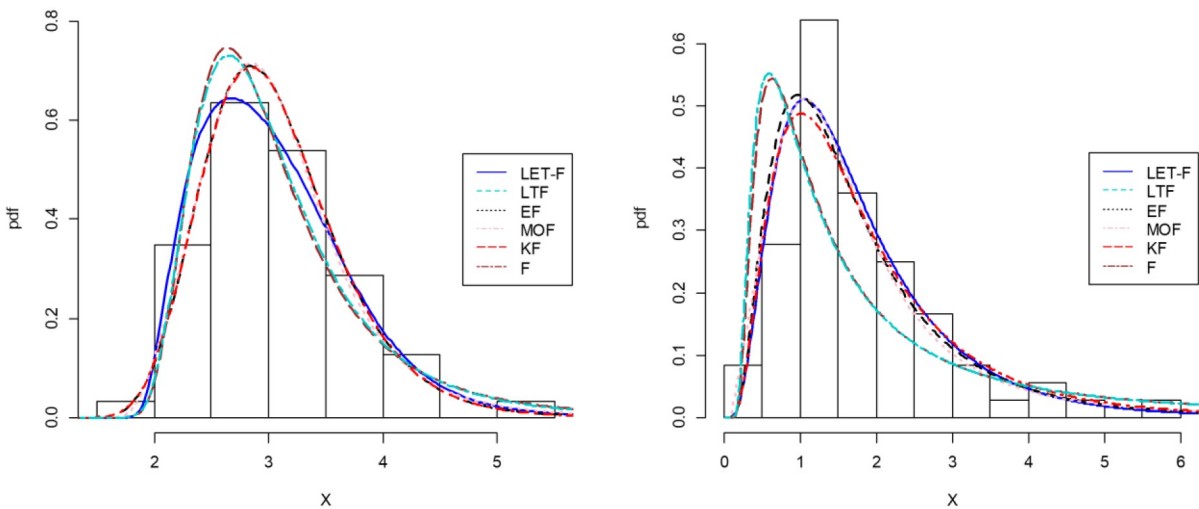

**Fig 4. Histogram and estimated pdf of the models for data set 1 (left) and data set 2 (right).**

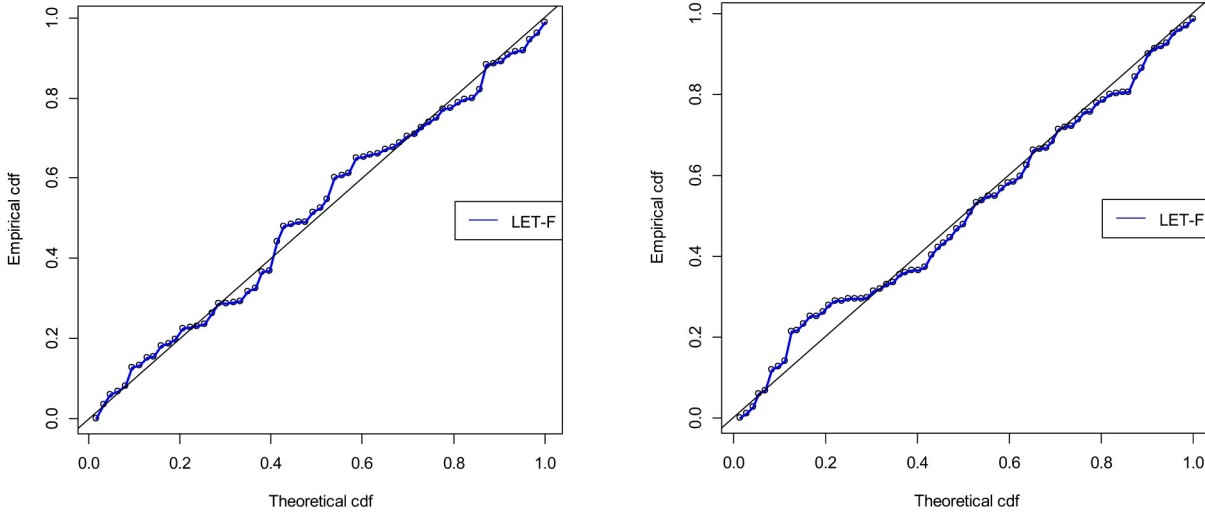

**Fig 5. PP-Plots of the *LET-F* model for data set 1 (left) and data set 2 (right).**

## Right-censored data

We now consider a data set presented by [22] and also used by [23]. The data is about the recurrence of leukemia of 46 patients (per year) who received autologous marrow. The full data set is given below where the plus sign indicates that observations are censored:

0.0301, 0.0384, 0.0630, 0.0849, 0.0877, 0.0959, 0.1397, 0.1616, 0.1699, 0.2137, 0.2137, 0.2164, 0.2384, 0.2712, 0.2740, 0.3863, 0.4384, 0.4548, 0.5918, 0.6000, 0.6438, 0.6849, 0.7397, 0.8575,

**Table 6.  *MLE*, its *SE* and *BE* with posterior *SD* of the considered models.**

| Model | Parameter | Data 1 | | Data 2 | |
|---|---|---|---|---|---|
| | | *MLE* | *BE* | *MLE* | *BE* |
| *LET-F* | $\hat{\lambda}$ | -0.6887 (0.0048) | 0.1011 (0.1836) | 51.9403 (81.9581) | 21.2228 (6.5014) |
| | $\hat{\alpha}$ | 9.8006 (1.3536) | 5.2812 (0.4522) | 0.4453 (0.1776) | 0.5755 (0.0657) |
| | $\hat{\beta}$ | 2.2339 (0.0744) | 2.7535 (0.0855) | 45.9901 (99.4069) | 14.3989 (3.6077) |
| *LTF* | $\hat{\beta}$ | 5.8853 (0.5330) | 5.7803 (0.5487) | 1.2654 (0.0884) | 1.2562 (0.0912) |
| | $\hat{\lambda}$ | 2.6235 (0.0618) | 2.6190 (0.0712) | 0.8600 (0.0894) | 0.8629 (0.0924) |
| *EF* | $\hat{\beta}$ | 2.4218 (1.6970) | 2.1103 (0.3117) | 0.6013 (0.0755) | 0.6046 (0.0576) |
| | $\hat{\lambda}$ | 4.2205 (2.4978) | 4.8781 (0.6147) | 8.5769 (3.8476) | 9.0086 (2.2129) |
| | $\hat{\alpha}$ | 6.6984 (13.027) | 6.3642 (3.8191) | 12.1029 (5.1417) | 12.9036 (3.3128) |
| *MOF* | $\hat{\beta}$ | 7.8946 (1.1419) | 6.8401 (0.2114) | 2.5532 (0.1991) | 1.8941 (0.15351) |
| | $\hat{\lambda}$ | 2.2055 (0.2335) | 2.3815 (1.0483) | 0.1762 (0.0285) | 0.3085 (0.0769) |
| | $\hat{\alpha}$ | 10.2274 (12.2381) | 5.4738 (3.5144) | 223.4801 (116.671) | 18.3103 (4.7330) |
| *KF* | $\hat{\alpha}$ | 9.4893 (20.363) | 2.7963 (3.2455) | 2.6846 (0.7402) | 5.3471 (2.3419) |
| | $\hat{\theta}$ | 7.0027 (13.815) | 5.4046 (2.4840) | 12.8647 (4.1109) | 8.1203 (2.7692) |
| | $\hat{\lambda}$ | 1.6622 (1.5250) | 3.7789 (1.7130) | 1.81258 (2.9768) | 0.7222 (0.7542) |
| | $\hat{\beta}$ | 2.3784 (1.6768) | 2.8077 (0.7167) | 0.5830 (0.0568) | 0.6691 (0.0882) |
| *F* | $\hat{\beta}$ | 5.4351 (0.5078) | 5.3630 (0.5342) | 1.1721 (0.0842) | 1.1620 (0.0855) |
| | $\hat{\lambda}$ | 2.7207 (0.0667) | 2.7202 (0.0791) | 1.0589 (0.1133) | 1.0617 (0.1142) |

**Table 7. The *MLE*, its *SE* and *BE* with posterior *SD* for different parameters, together with the log-likelihood (*L*), Akaike Information Criterion (*AIC*) and Deviance Information Criterion (*DIC*).**

| Distribution | Parameter | MLE | BE | L | AIC | DIC |
|---|---|---|---|---|---|---|
| LET-F | $\hat{\alpha}$ | 0.4430 (0.314) | 0.5156 (0.0714) | -45.52 | 97.03 | 94.661 |
| | $\hat{\beta}$ | 1.2010 (3.999) | 0.6475 (0.3851) | | | |
| | $\hat{\lambda}$ | 1.1990 (4.328) | 0.2161 (0.4856) | | | |
| LTF | $\hat{\alpha}$ | 0.6570 (0.1410) | 0.5958 (0.0792) | -45.33 | 96.66 | 94.837 |
| | $\hat{\lambda}$ | 0.3140 (0.1240) | 0.3612 (0.0993) | | | |
| | $\hat{p}$ | 0.1250 (0.1260) | 0.0483 (0.0613) | | | |
| LTW | $\hat{c}$ | 0.9012 (0.2117) | 0.8834 (0.1543) | -46.56 | 99.12 | 101.439 |
| | $\hat{\lambda}$ | 1.7857 (0.4495) | 0.8228 (0.3328) | | | |
| | $\hat{P}$ | 0.2721 (0.0676) | 0.2357 (0.0863) | | | |
| LTWL | $\hat{\alpha}$ | 0.9452 (0.1363) | 0.8855 (0.2218) | -46.15 | 98.3 | 100.434 |
| | $\hat{\lambda}$ | 0.6888 (0.1363) | 1.7482 (0.4483) | | | |
| | $\hat{P}$ | 0.2689 (0.0683) | 0.2614 (0.0669) | | | |

0.9096, 0.9644, 1.0082, 1.2822, 1.3452, 1.4000, 1.5260, 1.7205+, 1.9890+, 2.2438+, 2.5068+, 2.6466+, 3.0384, 3.1726+, 3.4411, 4.4219+, 4.4356+, 4.5863+, 4.6904+, 4.7808+, 4.9863+, 5.0000+.

This data set is random censored, see [22]. Here we compare our *LET-F* model with three models proposed recently by [22], namely, the long term Fréchet (*LTF*), the long term Weibull (*LTW*) and long term weighted Lindley (*LTWL*) distributions. The general form of a long term survival function is $S^*(x) = p + (1 − p) S(x)$, where $S(x)$ is the survival function of any distribution and $p$ denotes the probability of being cured. The corresponding distributions and pdfs can then be deduced from this mixture survival function.

This time, we use the Akaike information criterion (*AIC*) and the *DIC* as model comparison; the smaller its values, the better the fit (of course, the same tests as in the previous section can also be run here). Table 7 contains the maximum likelihood and Bayes estimates of the parameters. For quantification of variability of the estimates, *SEs* of the *MLEs* (in parenthesis) and *SDs* (in parenthesis) of the posterior distributions are reported. The log-likelihood (*L*) value and *AIC* of the proposed *LET-F* model are almost the same as those of the *LTF* model, and clearly smaller than for the other two models. While considering the *DIC* value, our proposed *LET-F* performs better than all other competitive models. Additionally, we present in Fig 6 the empirical survival function adjusted by the Kaplan-Meier estimator (*KME*) for our *LET-F* and the other three *LT* survival distributions.

## Conclusion

In this paper, we have proposed a new general construction of flexible lifetime distributions by rendering any existing baseline distribution more versatile through a simple transformation. We have discussed properties of the new models and explained how to estimate the parameters for complete and censored data sets. A Monte Carlo and hit-and-run Metropolis-Hasting simulations studies has revealed that the classical and Bayesian estimation procedures work well. On the basis of three distinct real data sets, we could see that the *LET-F* model, based on the Fréchet distribution as baseline distribution, is a very good competitor to existing distributions, especially to existing generalizations of the Fréchet. These good fitting capacities,

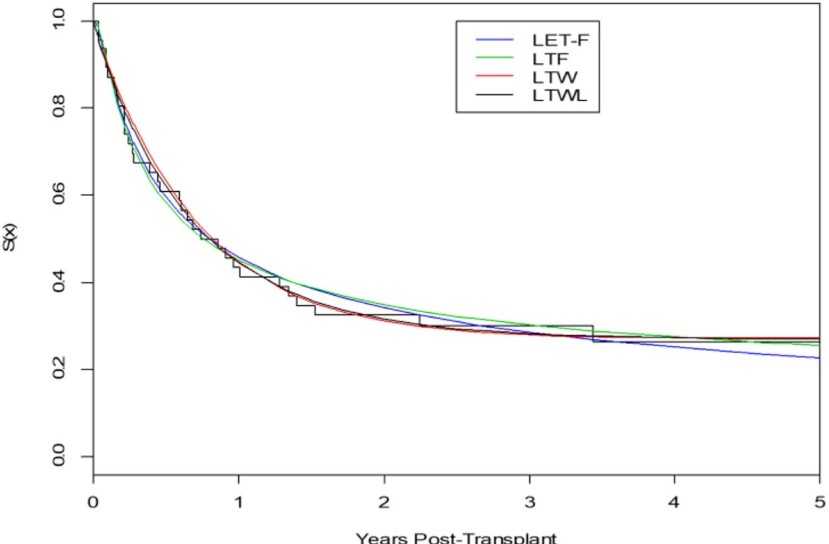

**Fig 6. Survival functions adjusted by *KME* for all considered models.**

combined with the simplicity of our proposal, make a strong case for using our construction in several practical situations.

## Supporting information

**S1 Appendix. Probability density functions of the competitors models and data sets.**
(DOCX)

**S1 Dataset.**
(DOCX)

## Author Contributions

**Conceptualization:** Zawar Hussain.

**Data curation:** Muhammad Aslam.

**Formal analysis:** Muhammad Aslam.

**Methodology:** Muhammad Aslam.

**Software:** Muhammad Aslam, Zawar Hussain.

**Writing – original draft:** Muhammad Aslam, Christophe Ley.

**Writing – review & editing:** Christophe Ley, Said Farooq Shah, Zahid Asghar.

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
