## [Decision Letter · Decision Letter 0]

11 Feb 2020

PONE-D-20-00826

A new generator for proposing flexible lifetime distributions and its properties

PLOS ONE

Dear Dr. Aslam,

Thank you for submitting your manuscript to PLOS ONE. After careful consideration, we feel that it has merit but does not fully meet PLOS ONE’s publication criteria as it currently stands. Therefore, we invite you to submit a revised version of the manuscript that addresses the points raised during the review process.

We would appreciate receiving your revised manuscript by Mar 27 2020 11:59PM. To enhance the reproducibility of your results, we recommend that if applicable you deposit your laboratory protocols in protocols.io, where a protocol can be assigned its own identifier (DOI) such that it can be cited independently in the future. For instructions see: http://journals.plos.org/plosone/s/submission-guidelines#loc-laboratory-protocols

We look forward to receiving your revised manuscript.

Kind regards,

Feng Chen

Academic Editor

PLOS ONE

Journal Requirements:

2. Please include a separate caption for each figure in your manuscript.

Reviewers' comments:

Reviewer's Responses to Questions

**Comments to the Author**

1. Is the manuscript technically sound, and do the data support the conclusions?

Reviewer #1: Yes

Reviewer #2: Partly

2. Has the statistical analysis been performed appropriately and rigorously? 

Reviewer #1: Yes

Reviewer #2: Yes

3. Have the authors made all data underlying the findings in their manuscript fully available?

Reviewer #1: Yes

Reviewer #2: Yes

4. Is the manuscript presented in an intelligible fashion and written in standard English?

Reviewer #1: Yes

Reviewer #2: Yes

5. Review Comments to the Author

Reviewer #1: This paper proposes a new generator for flexible lifetime distributions and discusses its statistical properties. Its performance is demonstrated by a Monte Carlo simulation study and three real-life applications. The paper is generally well structured and easy to access. A limitation is that the comparison between the proposed model and other alternatives only focuses on the goodness-of-fit and the results of AIC suggest that the difference seems to be insignificant. From the perspective of practical application, the model complexity should also be considered. The authors are suggested to conduct the model comparison using Bayesian methods. The deviance information criterion available in Bayesian inference, which is deemed as a generalization of the AIC, provides a combined measure of model fit and complexity. Please refer to some representative works which also model the censored continuous variables as in this paper, including:

A Bayesian spatial random parameters Tobit model for analyzing crash rates on roadway segments. Accident Analysis and Prevention, 2017, 100: 37-43.

A multivariate random parameters Tobit model for analyzing highway crash rate by injury severity. Accident Analysis and Prevention, 2017, 99: 184-191.

Incorporating temporal correlation into a multivariate random parameters Tobit model for modeling crash rate by injury severity. Transportmetrica A: Transport Science, 2018, 14 (3): 177-191.

Jointly modeling area-level crash rates by severity: A Bayesian multivariate random-parameters spatio-temporal Tobit regression. Transportmetrica A: Transport Science, 2019, 15(2): 1867-1884.

Reviewer #2: The topic of this paper is interesting and the methods sound. There are several suggestions to improve this paper.

1. “we will take the Fréchet as example” could be “we will take the Fréchet distribution as example”. And some references are needed.

2. “Typically this requires numerical optimization techniques such as Newton-Raphson.” References are needed for these techniques.

3. This paper lacks of references in the last decade.

4. “The score equations, and associated maximum likelihood estimates, are obtained along the same lines as in the previous sections, hence we leave this to the reader.” This sentence is not suitable. Some reference could be added instead of letting reader guess. For example, the following ones.

[1] Investigation on the Injury Severity of Drivers in Rear-End Collisions Between Cars Using a Random Parameters Bivariate Ordered Probit Model, International Journal of Environmental Research and Public Health, 2019, 16(14) , 2632.

[2] Injury severities of truck drivers in single- and multi-vehicle accidents on rural highway, Accident Analysis and Prevention, 2011, 43(5), 1677-1688.

[3] Analysis of hourly crash likelihood using unbalanced panel data mixed logit model and real-time driving environmental big data. 2018, JOURNAL OF SAFETY RESEARCH. 65: 153-159.

[4] Investigating the Differences of Single- and Multi-vehicle Accident Probability Using Mixed Logit Model, Journal of Advanced Transportation, 2018, UNSP 2702360.

5. For AIC, these references could also be referred to.

6. There are some ◇ in this paper, which might be not correct. For example, Page 10.

7. For censored data, Tobit models are frequently used, which could be referred to the following papers.

[5] Modeling crash rates for a mountainous highway using refined-scale panel data”, Transportation Research Record, 2015, 2515:10-16.

[6] Refined-scale panel data crash rate analysis using random-effects tobit model, Accident Analysis and Prevention, 2014, 73, 323-332.

8. “a p-value of 0.02663451” could be “a p-value of 0.027”.

6. PLOS authors have the option to publish the peer review history of their article (what does this mean?). If published, this will include your full peer review and any attached files.

Reviewer #1: No

Reviewer #2: No

---

## [Author Response · Author response to Decision Letter 0]

26 Mar 2020

Reviewer #1: This paper proposes a new generator for flexible lifetime distributions and discusses its statistical properties. Its performance is demonstrated by a Monte Carlo simulation study and three real-life applications. The paper is generally well structured and easy to access. A limitation is that the comparison between the proposed model and other alternatives only focuses on the goodness-of-fit and the results of AIC suggest that the difference seems to be insignificant. From the perspective of practical application, the model complexity should also be considered. The authors are suggested to conduct the model comparison using Bayesian methods. The deviance information criterion available in Bayesian inference, which is deemed as a generalization of the AIC, provides a combined measure of model fit and complexity. Please refer to some representative works which also model the censored continuous variables as in this paper, including: 

A Bayesian spatial random parameters Tobit model for analyzing crash rates on roadway segments. Accident Analysis and Prevention, 2017, 100: 37-43. 

A multivariate random parameters Tobit model for analyzing highway crash rate by injury severity. Accident Analysis and Prevention, 2017, 99: 184-191. 

Incorporating temporal correlation into a multivariate random parameters Tobit model for modeling crash rate by injury severity. Transportmetrica A: Transport Science, 2018, 14 (3): 177-191. 

Jointly modeling area-level crash rates by severity: A Bayesian multivariate random-parameters spatio-temporal Tobit regression. Transportmetrica A: Transport Science, 2019, 15(2): 1867-1884.

Answer: As per suggestion of the learned reviewer, we have calculated DIC and reported the results along with the results of AIC and other goodness of fit statistics for all three considered data sets. Moreover, Bayes estimates and posterior standard deviations are also reported. Also, some relevant papers reporting DIC as a goodness of fit criterion have been cited at appropriate places. 

Reviewer #2: The topic of this paper is interesting and the methods sound. There are several suggestions to improve this paper.

1. “we will take the Fréchet as example” could be “we will take the Fréchet distribution as example”. And some references are needed. 

Answer: corrected as suggested. Relevant reference articles have also been mentioned.

2. “Typically this requires numerical optimization techniques such as Newton-Raphson.” References are needed for these techniques.

Answer: References have been included.

3. This paper lacks of references in the last decade.

Answer: Yes! We agree with the referee. Unfortunately, we could not find any major transformer or technique related to this topic. Most of the recent work is based on extensions of previously proposed methods. We have included only major/ well known methods available in the literature. It is to be noted that only those frechet distributions which are generated from well known transformations are considered for comparison purposes using different data sets. As far as censoring schemes are concerned, we have included more recent work. 

4. “The score equations, and associated maximum likelihood estimates, are obtained along the same lines as in the previous sections, hence we leave this to the reader.” This sentence is not suitable. Some reference could be added instead of letting reader guess. For example, the following ones.

[1] Investigation on the Injury Severity of Drivers in Rear-End Collisions Between Cars Using a Random Parameters Bivariate Ordered Probit Model, International Journal of Environmental Research and Public Health, 2019, 16(14) , 2632.

[2] Injury severities of truck drivers in single- and multi-vehicle accidents on rural highway, Accident Analysis and Prevention, 2011, 43(5), 1677-1688.

[3] Analysis of hourly crash likelihood using unbalanced panel data mixed logit model and real-time driving environmental big data. 2018, JOURNAL OF SAFETY RESEARCH. 65: 153-159.

[4] Investigating the Differences of Single- and Multi-vehicle Accident Probability Using Mixed Logit Model, Journal of Advanced Transportation, 2018, UNSP 2702360.

Answer: Actually, this line was written for the sake of brevity. Obviously, score equations can be obtained by taking first derivative of log-likelihood function with respect to the parameters and then equating to zero. In earlier draft of the paper, we did not include these simple equations to save the space. To support the brevity, related reference papers have been included now in the revised manuscript. 

5. For AIC, these references could also be referred to.

Answer: The mentioned reference papers have been included.

6. There are some ◇ in this paper, which might be not correct. For example, Page 10.

Answer: corrected as mentioned.

7. For censored data, Tobit models are frequently used, which could be referred to the following papers.

[5] Modeling crash rates for a mountainous highway using refined-scale panel data”, Transportation Research Record, 2015, 2515:10-16.

[6] Refined-scale panel data crash rate analysis using random-effects tobit model, Accident Analysis and Prevention, 2014, 73, 323-332.

Answer: Relevant papers discussing tobit models have been cited. 

8. “a p-value of 0.02663451” could be “a p-value of 0.027”.

Answer: Changed by rounding all the results on p-values to four decimal places.

---

## [Decision Letter · Decision Letter 1]

3 Apr 2020

A new generator for proposing flexible lifetime distributions and its properties

PONE-D-20-00826R1

Dear Dr. Aslam,

We are pleased to inform you that your manuscript has been judged scientifically suitable for publication and will be formally accepted for publication once it complies with all outstanding technical requirements.

With kind regards,

Feng Chen

Academic Editor

PLOS ONE

Additional Editor Comments (optional):

Reviewers' comments:

Reviewer's Responses to Questions

**Comments to the Author**

1. If the authors have adequately addressed your comments raised in a previous round of review and you feel that this manuscript is now acceptable for publication, you may indicate that here to bypass the “Comments to the Author” section, enter your conflict of interest statement in the “Confidential to Editor” section, and submit your "Accept" recommendation.

Reviewer #1: All comments have been addressed

Reviewer #2: All comments have been addressed

2. Is the manuscript technically sound, and do the data support the conclusions?

Reviewer #1: (No Response)

Reviewer #2: Yes

3. Has the statistical analysis been performed appropriately and rigorously? 

Reviewer #1: (No Response)

Reviewer #2: Yes

4. Have the authors made all data underlying the findings in their manuscript fully available?

Reviewer #1: (No Response)

Reviewer #2: Yes

5. Is the manuscript presented in an intelligible fashion and written in standard English?

Reviewer #1: (No Response)

Reviewer #2: Yes

6. Review Comments to the Author

Reviewer #1: (No Response)

Reviewer #2: (No Response)

7. PLOS authors have the option to publish the peer review history of their article (what does this mean?). If published, this will include your full peer review and any attached files.

Reviewer #1: No

Reviewer #2: No

---

## [Editor Report · Acceptance letter]

17 Apr 2020

PONE-D-20-00826R1 

A new generator for proposing flexible lifetime distributions and its properties 

Dear Dr. Aslam:

I am pleased to inform you that your manuscript has been deemed suitable for publication in PLOS ONE. Congratulations! Your manuscript is now with our production department. 

With kind regards,

on behalf of

Dr. Feng Chen 

Academic Editor

PLOS ONE